# Sugar Beet Processing Wastewater Treatment by Microalgae through Biosorption

Nadiia Khakimova [1,*], Nikola Maravić [2], Petar Davidović [1], Dajana Blagojević [1], Milena Bečelić-Tomin [3], Jelica Simeunović [1], Vesna Pešić [3], Zita Šereš [2], Anamarija Mandić [4], Milica Pojić [4] and Aleksandra Mišan [4]

[1] Department of Biology and Ecology, Faculty of Sciences, University of Novi Sad, Trg Dositeja Obradovića 3, 21000 Novi Sad, Serbia; petar.davidovic@dbe.uns.ac.rs (P.D.); dajana.kovac@dbe.uns.ac.rs (D.B.); jelica.simeunovic@dbe.uns.ac.rs (J.S.)

[2] Faculty of Technology, University of Novi Sad, Bul. Cara Lazara 1, 21000 Novi Sad, Serbia; maravic@uns.ac.rs (N.M.); zitas@tf.uns.ac.rs (Z.Š.)

[3] Department of Chemistry, Biochemistry and Environmental Protection, Faculty of Sciences, University of Novi Sad, Trg Dositeja Obradovića 3, 21000 Novi Sad, Serbia; milena.becelic-tomin@dh.uns.ac.rs (M.B.-T.); vesna.pesic@dh.uns.ac.rs (V.P.)

[4] Institute of Food Technology in Novi Sad, University of Novi Sad, Bul. Cara Lazara 1, 21000 Novi Sad, Serbia; anamarija.mandic@fins.uns.ac.rs (A.M.); milica.pojic@fins.uns.ac.rs (M.P.); aleksandra.misan@fins.uns.ac.rs (A.M.)

* Correspondence: nadiia.khakimova@dbe.uns.ac.rs

**Abstract:** The aim of this study was to investigate the potential of environmental pollution reduction of sugar beet processing factory wastewater by the biorefinery approach and integration of microalgae biomass production. In the present study, *Chlorella vulgaris* was cultivated in wastewater collected from a sugar beet processing factory at the beginning and at the end of a sugar plant campaign in an aerobic bioreactor on a laboratory scale under controlled conditions, with an air flow of 0.4 L/min, a temperature of 26 °C, and pH = 8. Microalgae showed effective nutrient remediation from wastewater. During wastewater treatment, chemical oxygen demand (COD) and biological oxygen demand (BOD) removal efficiency was 93.7% and 98.1%, respectively; total organic carbon (TOC) content decreased by 95.7%. Nitrites and nitrates decreased by 96%, while the biggest decrease in metal ions was achieved for Ca and Mn (82.7% and 97.6%, respectively). The findings of this study suggest that coupling microalgae cultivation and wastewater treatment has a lot of potential for reducing contamination through biosorption, while also providing environmental advantages.

**Keywords:** wastewater treatment; microalgae cultivation; biosorption; nutrient removal; biorefinery concept

## 1. Introduction

The sugar industry, with global production of sugar exceeding 18 million tons annually, is one of the most important agro-based industries, in which sugar beet accounts for more than 20% of global sugar production [1].

As the European Union set the goal to reach carbon neutrality by 2050, the sugar industry has decreased its $CO_2$ emissions by 51% compared to 1990, but achieving climate neutrality still presents a real challenge for this sector. The generation of enormous amounts of pulp, the consumption of large quantities of lime (which are transformed into sludges), the production of vinasse, and high consumption of energy and water are the main sources of sustainability challenges and environmental management problems in traditional beet sugar processing [2].

Generally, the sugar beet industry is one of the top water users and wastewater producers, although water consumption depends on technological processes within the plant [3]. Even though modern wastewater treatment technologies provide more efficient water use, such as water reuse, regeneration, and recycling, in older factories, the consumption of

25–45 kg water per 100 kg beet and discharges of an even larger quantity of wastewater (including water contained in the beet processed) are still considered normal [4]. Processing wastewaters, if not properly managed, are a serious risk to human beings, the environment, and the recipient's aquatic life, as they contain a high concentration of organic compounds, especially soluble and insoluble polysaccharides, which presents an ideal environment for the proliferation of microbes [4].

Various physical, chemical, and biological methods for the treatment of sugar factory wastewater have been proposed aiming at the reduction of chemical oxygen demand (COD). Some advanced technologies involving anaerobic digestion are considered to be the preferred methods of wastewater treatment; however, these processes are unable to remove biological nitrogen and phosphorus, require frequent adjustments for alkalinity, and are yet to be feasible due to large land requirements, byproduct formation, and high operational costs [3].

Recently, concern has grown over the sustainability of conventional wastewater treatment systems in terms of economic feasibility and environmental impact given the fact that standards to improve water quality have become more stringent. This implies higher energy consumption and greenhouse gas emissions, aspects that have become key factors concerning the overall performance of wastewater treatment. It is estimated that annual $CO_2$ emissions from electricity consumed for wastewater treatment in Germany are 2.2 million tons, around 2.1 million tons in the United Kingdom, and approximately 11.5 million tons in the United States [5].

In recent years, the idea of integrating mixotrophic microalgae into wastewater treatment has received much attention given the fact that the use of microalgae in wastewater treatment is a cost-effective and feasible method for biofixation of $CO_2$ [6]. Apart from their ability to utilize organic and inorganic C, N, and P for their growth, the principal advantage of incorporating microalgae into wastewater treatment is the generation of $O_2$ through photosynthesis, necessary for heterotrophic bacteria to biodegrade carbonaceous materials [7]. It is already known that multiple factors such as light, pH, nitrogen-to-phosphorus ratio, temperature, and carbon source and bacteria concentration influence algal productivity, so it is difficult to compare the effect of algal culture in processing wastewater treatment [8–11]. In addition to removing pollutants, the cultivation of microalgae in conjunction with wastewater treatment can provide lipids that can be converted into biodiesel [8,12].

However, several practical and economic challenges still hinder the implementation of microalgae to treat wastewater on a large scale. In a number of papers, wastewater pretreatment, such as dilution, nutrient addition, and anaerobic digestion, was necessary before microalgal inoculation [8–11]. Among others, the challenge we tried to address in this work is harvesting, which constitutes one of the main technoeconomic limitations of this technology [5].

The aim of this study was to investigate the potential of environmental pollution reduction of a sugar beet processing factory wastewater by the biorefinery approach and integration of microalgae biomass production. Thus, the following objectives were set: (1) selection of microalgae strains capable of growing in sugar beet processing wastewater, (2) testing the ability of selected microalgae to tolerate the fluctuating composition of wastewater, and (3) to propose a microalgae harvesting method.

## 2. Materials and Methods

### 2.1. Wastewater Sampling and Pretreatment

In this case study, wastewater from a sugar plant in Crvenka (Vojvodina, Serbia) was tested. The functional scheme of the current sugar plant operation and wastewater discharge upon production is presented in Figure 1. Sugar beet processing plants are operated only for approximately 100 days/year with a fluctuating characteristic of wastewater mostly depending on the weather conditions and corresponding sugar beet quality. The sugar factory in Crvenka processed 770,000 tons of sugar beets and produced 750,000 m$^3$ of

wastewater in 118 working days. According to [13], a significantly higher wastewater load is expected towards the end of the sugar beet processing season followed by rainfall, frost, and deteriorated sugar beet.

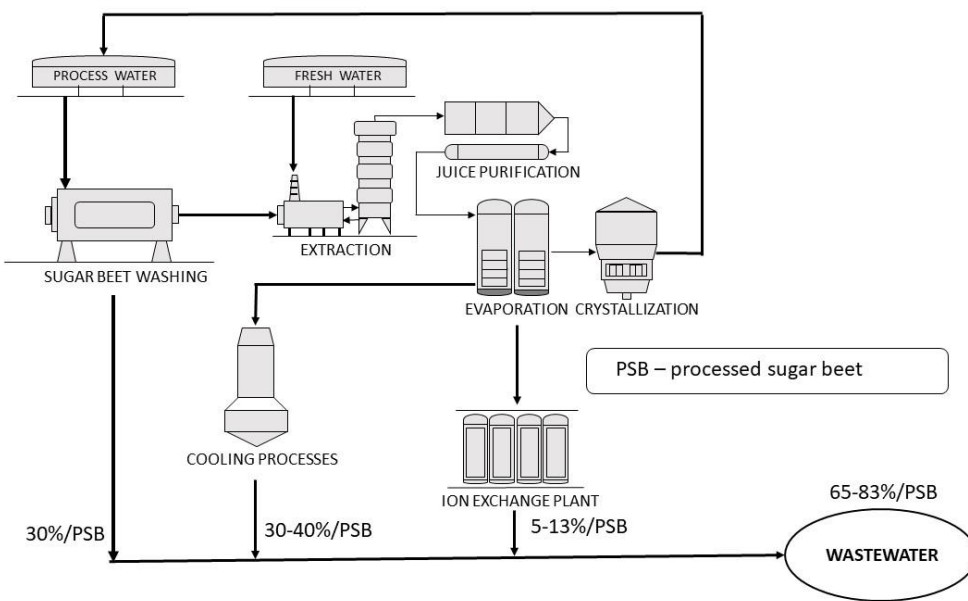

**Figure 1.** Production scheme of the sugar plant. In the process, each stream with PSB equals to %water per processed sugar beet.

Wastewater samples were obtained in 2019 for the preliminary experiment and in 2020 for the main experiment. Sugar plant wastewater was not pretreated or diluted before sampling. The wastewater was coarsely filtered using a gauze filter in the laboratory to remove large solid materials such as rags, wood, and heavier grit particles to avoid damage to the bioreactor. Samples were transferred to 5 L plastic bottles and stored at −10 °C until the experiments were performed.

*2.2. Selecting of Microalgae Strains and Cultivation Conditions*

2.2.1. Selection of Most Adaptive Microalgae Strain

Microalgal strains of *Selenastrum, Chlorella*, and *Nostoc* genera were pretested in order to select specific microorganisms for the corresponding sugar plant wastewater sample. Strains were taken from the Novi Sad culture collection of cyanobacteria and microalgae (NSCCC) at the Department of Biology and Ecology at the Faculty of Sciences, University of Novi Sad (UNSPMF). Cultures were grown in algae broth medium with 12 h light and dark cycles and T = 22–24 °C. All three strains were then cultivated under mixotrophic growth in raw and diluted (1:10) sugar plant wastewater sampled at the beginning of the sugar beet processing season. Growth was monitored in 250 mL glass bottles for 1 week. After cultivation and evaluation of cell growth and biomass production, *Chlorella vulgaris* was selected as the most stable and adaptive, with a higher growth rate in the undiluted wastewater sample compared to other species (results not presented).

2.2.2. Cultivation Condition in Preliminary Experiment

Cultivation was performed in a W11 Armfield Aerobic Digester—a lab-scale bioreactor. Addition of 8000 mL of undiluted sugar plant wastewater, sampled at the beginning of the sugar beet processing season, and 80 mL of suspended *C. vulgaris* culture with cell density 30,000 cells/mL, provided a predetermined ratio between wastewater and biomass of 1:100. The determination of *C. vulgaris* cell density was performed by direct cell counting using a

hemocytometer and the Olympus BX51 Fluorescence Microscope and expressed as number of cells per mL [14,15].

$$\text{Number of cells/mL} = (\text{number of cells/5}) \times 10^4 \tag{1}$$

Approximate cell size (in μm) was determined using special computer software.

During the course of the experiment, the constant air flow was set to 0.4 L/min, temperature was maintained at 26 °C, and pH was adjusted to pH = 8 at the start of the experiment, as suggested in the literature for optimal microalgae growth [16,17]. The period of microalgal cultivation was 6 weeks.

### 2.2.3. Biofilm-Based Microalgal Cultivation in the Main Experiment

Cultivation was performed in undiluted sugar plant wastewater, sampled two months after the processing season had started using the same lab-scale bioreactor and other experimental conditions such as in the preliminary experiment. The difference was the addition of a transparent polypropylene net, which was installed along lengthwise bioreactor walls to address the problem of microalgae harvesting.

### 2.3. Anaerobic and Aerobic Controls in the Main Experiment

The main experiment included two separate control sample bottles containing 8000 mL of undiluted sugar plant wastewater, sampled two months after the processing season had started without microalgae, one open with aeration (constant air flow was the identical as in the bioreactor—0.4 L/min) and the other closed without aeration. The temperature was maintained at 26 °C, and the initial pH = 5.6 was not corrected.

### 2.4. Chemical Characterization of Wastewater

Total suspended solids and dry residue were determined gravimetrically after drying and ashing at the temperature of $103 \pm 2$ °C and at $550 \pm 10$ °C, respectively [18].

Dissolved oxygen as the amount of molecular oxygen dissolved in water (in mg $O_2$/L and %) was measured daily directly using the $DO_2$ meter integrated with the W11 Armfield Aerobic Digester. pH values were measured daily using the C1020 Consort (Brussels, Belgium) pH meter. As the initial pH in the wastewater during both the preliminary and main experiment (Tables 1 and 2, respectively) was about 5.5, 8 mL of 1M NaOH was added to adjust the pH to 8, as it was proven that the optimum pH range for most microalgae growth is between 7 and 9 [9–11].

**Table 1.** Physicochemical characterization and removal efficiency of certain components in the initial sugar plant wastewater and the wastewater after microalgae treatment during the preliminary experiment.

| Parameter | Units | Preliminary Experiment | | |
| :---: | :---: | :---: | :---: | :---: |
| | | Initial | Final | Removal Efficiency, % |
| pH | – | 5.6 | 9.3 | – |
| Total P | mg P/L | $0.104 \pm 0.002$ | – | – |
| Orthophosphates | mg P/L | $0.024 \pm 0$ | $0.018 \pm 0$ | 25 |
| Nitrite | mg N/L | <0.005 | – | – |
| Nitrate | mg N/L | $0.034 \pm 0$ | – | – |
| TKN | mg N/L | $42.2 \pm 1.83$ | $3.03 \pm 0.02$ | 92.8 |
| COD | mg $O_2$/L | $1046.4 \pm 10.39$ | $62 \pm 0.7$ | 94.07 |
| BOD | mg $O_2$/L | $715.5 \pm 4.9$ | <4 | 99.4 |
| TOC | mg C/l | – | – | – |
| Suspended solid residue | mg/L | $184.5 \pm 6.36$ | $15 \pm 0.7$ | 91.8 |
| Dry residue | mg/L | $1370.5 \pm 13.4$ | $974.5 \pm 3.18$ | 28.8 |
| Ash | mg/L | $897.5 \pm 4.94$ | $312 \pm 1.41$ | 65.2 |

Values are expressed as mean $\pm$ standard deviation ($n$ = 2).

**Table 2.** Physicochemical characterization and removal efficiency of certain components in the initial sugar plant wastewater and the wastewater after microalgae treatment during the main experiment.

| Parameter | Units | Experimental Samples | | | Control Samples | |
|---|---|---|---|---|---|---|
| | | Initial | Final | Removal Efficiency, % | Aerobic Control after 6 Weeks (without Algae) | Anaerobic Control after 6 Weeks (without Algae) |
| pH | – | 6.63 | 9.4 | – | – | – |
| Total P | mg P/L | 1.13 ± 0.01 | 0.547 ± 0.003 | 51.6 | 0.7 ± 0.008 | 0.68 ± 0.008 |
| orthophosphates | mg P/L | 1.1 ± 0.01 | 0.141 ± 0.004 | 87.2 | 0.6 ± 0.04 | 0.29 ± 0.002 |
| Nitrite | mg N/L | 0.15 ± 0.07 | 0.005 ± 0 | 96.7 | 0.065 ± 0.001 | 0.1415 ± 0.003 |
| Nitrate | mg N/L | 1.59 ± 0.01 | 0.063 ± 0 | 96.0 | <0.02 | <0.02 |
| TKN | mg N/L | 141 ± 1.4 | 62.05 ± 4.31 | 56.0 | 121 ± 2.82 | 125.5 ± 0.7 |
| COD | mg $O_2$/L | 8613 ± 5.0 | 541.5 ± 2.12 | 93.7 | 982 ± 4.24 | 5516 ± 24.04 |
| BOD | mg $O_2$/L | 4922 ± 3.5 | 93 ± 4.24 | 98.1 | 107.5 ± 4.9 | 5116 ± 22.6 |
| TOC | mg C/l | 4461 ± 1.41 | 192 ± 2.83 | 95.7 | – | – |
| Suspended solid Residue | mg/L | 413 ± 2.1 | <12 | 97.08 | 1757 ± 14.1 | 376.5 ± 4.9 |
| Dry residue | mg/L | 15303 ± 35.5 | 1744 ± 23.3 | – | 708 ± 16.9 | 5865 ± 101.8 |
| Ash | mg/L | 2240 ± 37.4 | 1285 ± 32.5 | – | 423 ± 4.24 | 1119 ± 16.9 |

Values are expressed as mean ± standard deviation (*n* = 2).

The determination of chemical oxygen demand and biological oxygen demand was performed according to Serbian ISO 6060:1994 and H1.002 [18]. Measurements were performed in duplicates. COD is the mass concentration of oxygen equivalent to the amount of dichromate consumed by dissolved and suspended matter when water samples are treated with this oxidant under specified conditions. Part of the test sample is refluxed in the presence of mercury (II) sulfate with a known amount of potassium dichromate and silver-based catalysts in strong sulfuric acid over a period of time, during which part of the dichromate is reduced by the oxidizable material present. The remaining dichromate is titrated with ammonium iron (II) sulfate. COD value is calculated from the amount of reduced dichromate. A 1 mol amount of dichromate is equivalent to 1.5 mol of oxygen.

For biological oxygen demand (BOD) measurement, 100 mL of wastewater was diluted with tap water and placed in special BOD bottles. Blanks were prepared, containing only distilled water and nitrification inhibitors. Microorganisms present or intentionally inoculated in a sample of water, which contains biodegradable organic substances, use oxygen for biochemical processes, and produce an equivalent amount of carbon dioxide. The process takes place in a closed system, where carbon dioxide is adsorbed by a strong base (KOH), and a progressive reduction in system pressure is measured. The BOD sensor is placed directly on the bottle and uses a microprocessor to store 5 BOD values measured at 24 h intervals. The current BOD value can be read at any time on the display, as well as after 5 days. The blank test (deionized water) is treated in the same way. The value of the blank is subtracted from the value of the sample.

Total carbon, as well as total organic carbon, was measured in duplicates by means of the TOC-V CPH/CPN Total Organic Carbon Analyzer according to Serbian ISO 8245:2007 and the Thermo Scientific TN 3000 Nitrogen Analyzer [19]. Total phosphorus was measured photometrically by producing molybdophosphoric acid from the interaction of orthophosphate ions in sulfuric acid solution with molybdate ions. The chemical is then converted to phosphomolybdenum blue by ascorbic acid [20].

Nitrates, nitrites, and ammonium ion ($NH_4^+$) were determined spectrophotometrically. The yellow compound formed by the reaction of sulfosalicylic acid (formed by adding sodium salicylate and sulfuric acid to the sample) with nitrates is measured spectrophotometrically and then treated with alkali. Ethylenediaminetetraacetic acid (EDTA) is added with alkali to prevent the deposition of potassium and magnesium salts. Sodium azide (sulfamic acid) is added to eliminate nitrite interference [20]. Nitrite ions in acid solution form with the sulfanilic acid, a diazonium salt that reacts with the N-(1-naftil)-

ethylenediamine dihydrochloride, giving a reddish violet azo dye, which was determined photometrically [21]. Ammonium ion ($NH_4^+$) in a strong alkaline solution is present entirely as ammonia. This reacts with hypochlorite ions to form monochloramine, which in turn reacts with 2-chlorophenol or thymol to form indophenol blue. This was then determined photometrically at $690-712$ nm [22]. Measurements were performed in duplicates.

In this study, total Kjeldahl nitrogen (TKN) was determined as the sum of ammonia-nitrogen and organically bound nitrogen; however, nitrate-nitrogen and nitrite-nitrogen are not included in TKN. Therefore, nitrates and nitrites were determined separately. Furthermore, previous studies have stated that determination of removal efficiency based only on NH4+ might give deceitful data, as a significant amount of NH3 is lost at pH higher than 9 [23,24].

Removal efficiency (RE) was calculated using the following equation:

$$RE(\%) = [(xi - xf))/xi] \times 100 \qquad (2)$$

where xi is the parameter value before the experiment, and xf is the parameter value after the experiment.

### 2.5. Metal Analysis in the Initial Wastewater and Wastewater after Algae Cultivation

After ash determination analysis in wastewater, the ash was collected and quantitatively transferred to a 50 mL flask. Ash was dissolved with concentrated nitric acid, and the flask was filled with distilled water. For the calibration curve, five calibration standards for each metal (Ca, Na, Cu, Pb, Cd, Fe, Mn, Mg, Zn, and K) and blanks were prepared, and the procedure of metal determination was carried out according to ISO 6869:2000 using the Varian, SpectrAA—10 Atomic Absorption Spectrometer [25].

## 3. Results

### 3.1. Preliminary Experiment

During the preliminary experiment, wastewater was taken at the beginning of the sugar plant campaign, so the organic matter content in the wastewater prior to microalgae treatment was significantly lower compared to the main experiment (Table 2), where wastewater was taken several months after the campaign had started. The physicochemical characterization of wastewater and nutrient removal efficiency in the preliminary experiment are presented in Table 1.

Removal Efficiency of Microalgae in the Preliminary Experiment

In the present study, mixotrophic growth was chosen for algal cultivation (Figure 2).

It is known that initial wastewater composition considerably impacts microalgal treatment efficiency. During their growth, microalgae require three essential elements: carbon, nitrogen, and phosphorus [8]. Performed chemical analysis showed that all of the necessary elements were initially present in the sugar plant wastewater (Table 1). No dilution was made to the initial wastewater in order to evaluate the possibility of using microalgae in primary wastewater treatment.

The efficiency of wastewater treatment by microalgae was estimated by the removal efficiency of TKN, COD, BOD, orthophosphates, and suspended solid residues. Generally, microalgae are able to consume nitrogen from organic (e.g., urea, nucleosides, purines, and amino acids) and inorganic (e.g., $NH_4^+$, $NO_3^-$, and $NO_2^-$) sources [7].

During the preliminary experiment, the amount of TKN in the initial wastewater was $42.2 \pm 1.83$ mg-N/L, and after the microalgae treatment, it dropped to $3.03 \pm 0.02$, indicating 92.8% of the removal efficiency for TKN. The initial concentration of nitrates and nitrites in the sugar plant wastewater was $0.034 \pm 0$ mg N/L and <0.005 mg N/L, respectively. The removal efficiency of nitrates and nitrites was not evaluated during the first experiment, as at the beginning of plant operation, wastewater was not that contaminated compared to the main experiment (Table 2), so during microalgae treatment, the $NO_3^-$ and $NO_2^-$ concentration had fallen below the level of detection. Only 25%

of orthophosphate removal was achieved during wastewater remediation. Microalgae cultivation also resulted in 94.07% and 99.4% removal rates for COD and BOD, respectively. The removal rate for total suspended solids accounted for 91.8%. Despite microalgae being used as the primary wastewater treatment, such removal rates are comparable with previous results, where similar efficiency was achieved while microalgae were used as tertiary to quinary wastewater treatments [5].

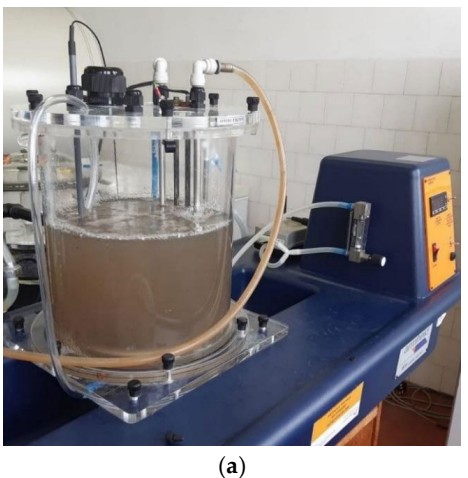
(**a**)

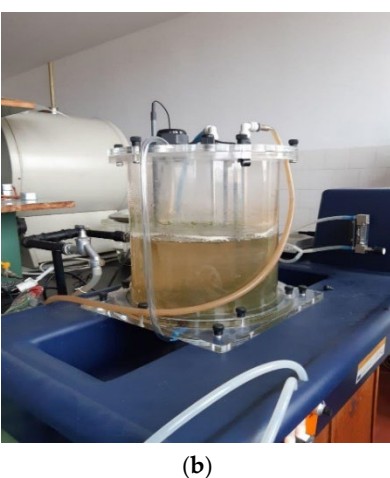
(**b**)

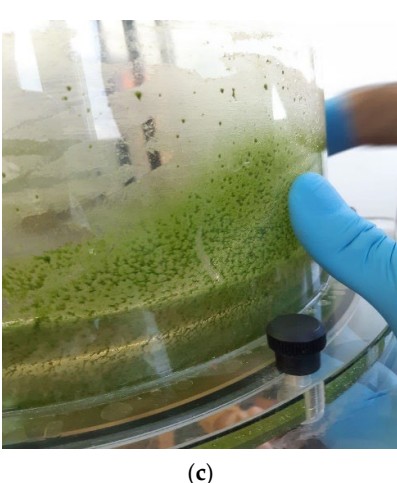
(**c**)

**Figure 2.** Photo of bioreactor during wastewater treatment using *Chlorella vulgaris* at the time of the preliminary experiment. (**a**) Wastewater in the bioreactor on the 1st day of microalgae inoculation. (**b**) Wastewater in the bioreactor after 4 weeks of treatment. (**c**). Biomass accumulation on the walls of bioreactor.

### 3.2. The Main Experiment

Characteristics of Sugar Plant Wastewater in the Main Experiment

As the preliminary experiment showed that microalgae can be effectively used for nutrient removal from sugar plant wastewater, demonstrating wastewater remediation with high removal rates and biomass accumulation, it was decided to proceed with the experiment, making certain modifications. During the main experiment, wastewater was taken from the same sugar plant but later during the campaign, which explains the significantly higher load of organic matter (Table 2).

In the preliminary experiment, a considerable amount of biomass accumulated on the walls of the bioreactor, and thus, quantitative microalgae harvesting was aggravated (Figure 2c). Due to the small cell size, density, and negative charge, it requires a significant amount of energy and relatively expensive methods to separate microalgal biomass from wastewater at the industrial scale cultivation [26].

Referring to the literature, during cultivation, microalgae cells develop extracellular polymeric substances that aid in the formation of biofilms when inoculated into porous materials such as a filter membrane or other cotton fibers [27]. Pollutants could be allowed to remain for a set period of time by using various membrane materials and changing the angle of the biofilm-attached reactor. The membrane materials soaked in the culture medium supply microalgae cells with nutrients and water [28]. Therefore, in the main experiment, net installation during microalgal cultivation was tested in order to address the problem of microalgae harvesting (Figure 3).

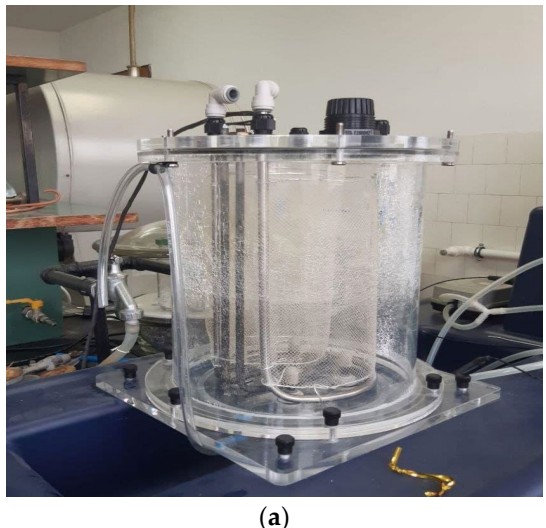
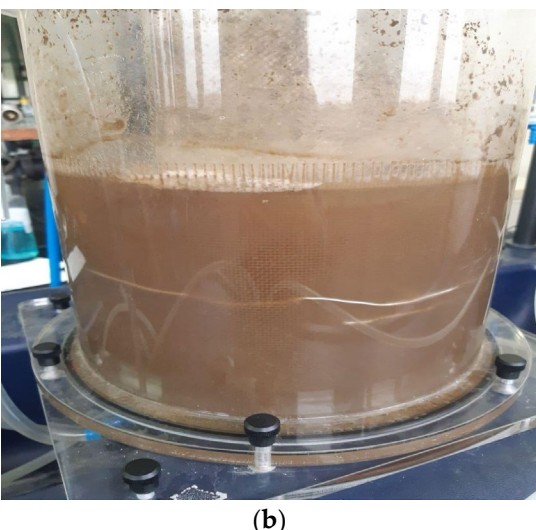

**(a)**                    **(b)**

**Figure 3.** (**a**) Net installation in the bioreactor before the wastewater treatment during the main experiment. (**b**) Wastewater in the bioreactor on the 1st day of treatment during the main experiment.

Moreover, it was decided to monitor removal rates more precisely, with weekly measurements of the main chemical parameters, in order to track purification dynamics.

In addition to removal efficiency, in order to separately evaluate microalgae wastewater treatment capacities, two controls, one with aeration and another without aeration, were set. Overall, both controls showed significantly smaller removal rates of organic matter in wastewater compared to microalgae treatment (Table 2).

*3.3. Removal Efficiency of Microalgae in the Main Experiment*

3.3.1. pH and Oxygen Content during Wastewater Treatment

Changes in pH values were measured daily (Figure 4a). During the main experiment, the pH increased from 8 to 9.3 after 38 cultivation days, with the biggest increase between Days 1 and 6, as well as between Days 11 and 21.

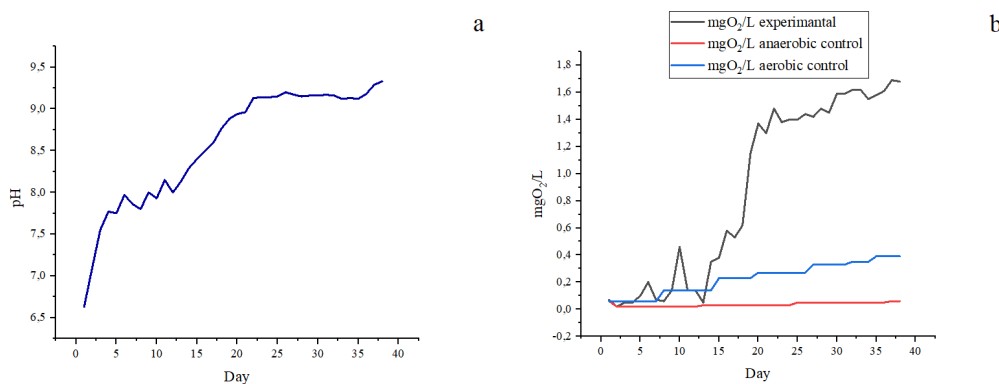

**Figure 4.** pH values during wastewater treatment in the main experiment (**a**). Wastewater oxygen content during microalgae cultivation in the bioreactor (black), anaerobic (orange), and aerobic (blue) control (**b**).

Oxygen content in the wastewater during the experiment was monitored daily and presented as mg $O_2$/L (Figure 4b). Overall, the level of oxygen increased during the whole cultivation period, from 0.007 mg $O_2$/L (0.8%—$O_2$) to 1.68 mg $O_2$/L (19.1%—$O_2$), with the steepest rise between Days 11 and 21, which directly correlates with the elevation of the pH value.

### 3.3.2. Nitrogen and Phosphorus Removal

In the main experiment, the TKN content in the sugar plant wastewater before microalgae inoculation was $141 \pm 1.4$ mg-N/L, and after 6 weeks of treatment, it decreased to $62.05 \pm 4.31$ mg-N/L, which indicated 56% of total nitrogen removal efficiency. Both nitrates and nitrites decreased significantly after cultivation compared to the initial level (Table 2, Figure 5). Nitrates decreased from $1.59 \pm 0.01$ mg-N/L to $0.063 \pm 0$ mg-N/L, which indicates 96.0% removal efficiency, while the concentration of nitrites in the wastewater dropped from $0.15 \pm 0.07$ mg-N/L to $0.005 \pm 0$ mg-N/L or 96.7% removal efficiency after microalgae treatment. The most intensive decline in nitrates and nitrites concentration occurred during the first 2 weeks of algae cultivation.

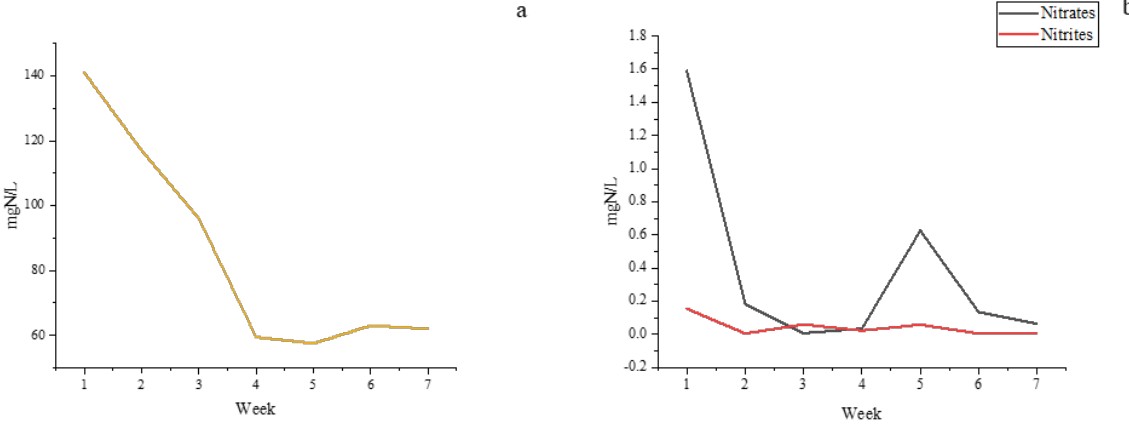

**Figure 5.** Changes in TKN (**a**), nitrate, and nitrite (**b**) concentration during wastewater treatment. Phosphorus and orthophosphate removal rates during the main experiment were higher compared to the preliminary experiment with values of 51.6% and 87.5%, respectively.

### 3.3.3. Metal Ion Removal from Wastewater

Metal ion content in sugar plant wastewater was analyzed at the beginning of the preliminary experiment, as well as at the beginning and at the end of the main experiment; moreover, the microalgal metal removal efficiency during the main experiment was evaluated (Table 3).

**Table 3.** Content of metal ions in wastewater with removal efficiency rates during the main experiment.

| Metal | Preliminary Experiment Initial Wastewater (mg/L) | Main Experiment Initial Wastewater (mg/L) | Main Experiment Wastewater after Algae Cultivation (mg/L) | Main Experiment Removal Efficiency, % |
|---|---|---|---|---|
| Ca | 14.95 | 16.62 | 2.88 | 82.7 |
| Na | 7.16 | 3.24 | 26.20 | |
| Cu | 0.005 | 0.0033 | 0.0028 | 15.1 |
| Pb | – | – | – | – |
| Cd | – | – | – | – |
| Fe | 0.43 | 0.30 | 0.21 | 30 |
| Mn | 0.07 | 0.17 | 0.004 | 97.6 |
| Mg | 2.72 | 2.93 | 2.74 | 6.5 |
| Zn | 0.06 | 0.009 | 0.01 | – |
| K | 1.4 | 4.05 | 6.37 | – |

In general, several metals were found in the wastewater in different concentrations, namely Ca, Na, Cu, Fe, Mn, Mg, Z, and K. Wastewater from the preliminary experiment sampled at the outset of sugar beet processing season had lower levels of Ca, Mn, Mg, and K but higher levels of Na, Cu, Fe, and Zn compared to the wastewater analyzed in the main experiment, two months after the processing season started. Removal efficiency by *C. vulgaris* for Ca, Cu, Fe, Mn, and Mg ranged from 6.5% to 97.6% and was the highest for Mn (97.6%) and Ca (82.7%) while lowest for Mg (6.5%).

### 3.3.4. COD, BOD, and TOC Removal

The efficiency of wastewater treatment by using microalgae was also evaluated by analyzing COD, BOD and TOC removal rates. The results are presented in Figure 6 and in Table 2.

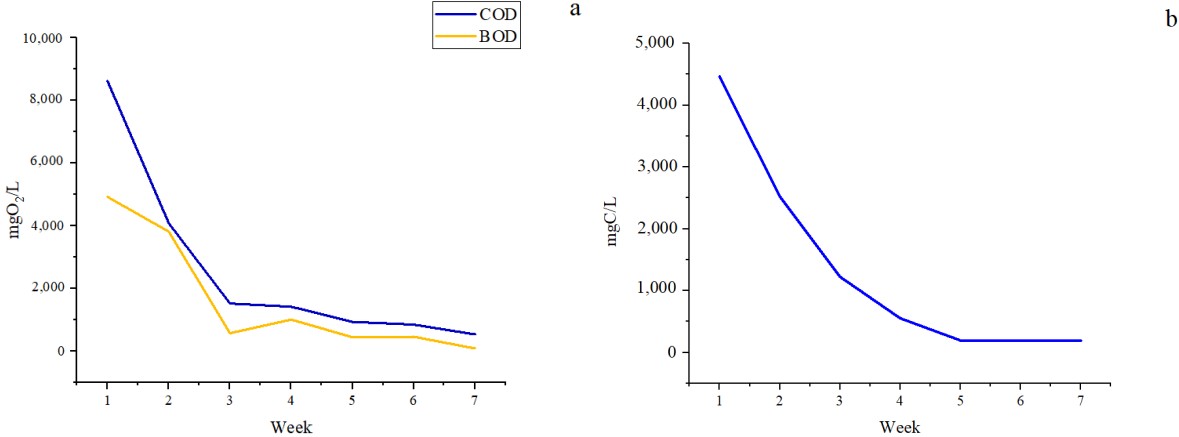

**Figure 6.** Changes in COD, BOD (**a**), and TOC (**b**) concentration during microalgae wastewater treatment.

Overall, COD, BOD, and TOC removal rates after the main experiment were 93.7%, 98.1%, and 95.7% respectively. However, the initial concentration level of COD and BOD were significantly higher during the main experiment compared to the preliminary experiment, as previously explained by the different sampling time in the sugar beet processing season. The COD initial concentration in the wastewater was 8 times higher in the main experiment compared to the preliminary experiment (Tables 1 and 2, respectively): $8613 \pm 5.0$ mg $O_2$/L and $1046.4 \pm 10.39$ mg $O_2$/L. While BOD concentration was approximately 6 times higher ($4922 \pm 3.5$ in the preliminary and $715.5 \pm 4.9$ in the main experiment).

### 4. Discussion

The results of the present study confirm that microalgae can effectively utilize nutrients from sugar plant wastewater and assist in bioremediation. Mixotrophic growth was chosen for microalgal cultivation, since it was previously stated as the most reliable and efficient in wastewater treatment, due to the fact that the mixotrophic type of cultivation overcomes the limitation of light requirement present during wastewater treatment, in contrast to the photoautotrophic nutrition mode [29]. During mixotrophic cultivation, microalgae could simultaneously use inorganic (for instance, $CO_2$) and organic compounds as carbon sources [30]. As a result, microalgae grown in a mixotrophic system synthesize compounds that are typical for both autotrophic (photosynthetic) and heterotrophic metabolisms at large rates. In addition, mixotrophic cultivation has been related to lower energy costs as compared to photoautotrophic cultivation, owing to its lower light intensity requirements [31]. Both control experiments did not show comparable removal rates: aerobic control showed a significant decrease in COD, BOD, and nitrates, while anaerobic treatment showed a noticeable decline in orthophosphates and suspended solids. These reactions are contingent on oxidation–reduction processes carried out by a variety of bacteria and fungi that are abundant in wastewaters [32].

It can be assumed that changes in pH values during the main experiment were the result of inorganic carbon assimilation by microalgae, as well as nitrogen consumption. As previously reported, increased ammonia volatilization and nitrate ions absorption could lead to higher pH values. However, if ammonia becomes the main nitrogen source for microalgae, it may lead to a significant pH decrease and inhibition of microalgae growth due to an excessively acidic environment [33–35]. Hawrot-Paw et al. [36] reported a pH range

from 7.98 to 8.54 during aquaculture wastewater treatment with *Chlorella minutissima* over a span of 10 days. Eze et al. [23] also described a significant rise in pH values from 8.0–8.5 to 10.5 after 28 days during the experimental purification of wastewater sampled from a wastewater treatment plant with *Desmodesmus* sp. Moreover, it was proven that the $CO_3^{2-}$ forms of inorganic carbon predominates in solutions with a pH above 10, and microalgae cannot fully utilize this form, which leads to a decrease in their biomass production and overall nutrient removal rates [37]. As the process was mixotrophic (i.e., both light as inorganic and nutrients as an organic source of carbon were used by microalgae for growth), microalgae were produced and released oxygen by the process of photosynthesis in the presence of daylight, which led to a rise in dissolved oxygen levels [35]. The corresponding conclusion is also in agreement with previous studies, where the correlation between the rise in dissolved oxygen levels and the increase in microalgae biomass was reported during microalgae cultivation in wastewater [38–40]. In contrast, control treatments without microalgae showed no (anaerobic) or very little (aerobic) dissolved oxygen level increase compared to the main treatment. Such results indicate that microalgae are essential for wastewater saturation with oxygen, hence stimulating the nitrification process by oxygenic photosynthesis [39]. Both pH and dissolved oxygen levels are important markers for monitoring and characterizing microalgae growth patterns.

The sharp increase in concentration of nitrates from Week 3 to 4 could be explained by the process of oxidation of ammonium nitrogen provoked by bacterial nitrification, and a similar effect was previously reported by Eze et al. [23]. Moreover, they reported 62% of total nitrogen removal efficiency after 28 days of wastewater cultivation using *Desmodesmus* sp.; however, the actual nitrogen removal efficiency by microalgae accounted only 48% out of 62%, as material N/P balance predicted that around a 14% loss of the initial ammonium nitrogen occurred due to $NH_3$ volatilization. Additionally, Aslan and Kapdan [41] reported that during algal treatment of synthetic wastewater, the $NH_4$-N removal rate was around 50% in water with an ammonia concentration of 41.8–92.8 mg-N/L, and it dropped to around 24% in wastewater samples where the $NH_4$-N concentration was above 129 mg-N/L. McGaughy et al. [42] reported a drop of nitrate levels in wastewater produced from hydrothermally treated septage during microalgae treatment with *Chlorella* sp., from $9.3 \pm 1.3$ to $5.2 \pm 0.2$ and then $1.7 \pm 0.2$ mg-N/L at Days 0, 5, and 10, respectively. The authors concluded that microalgae initially consume ammonia and other TKNs, while nitrates and nitrites are consumed secondarily. Therefore, nitrate- and nitrite-containing wastewaters require longer periods of time to be efficiently treated with microalgae. It is known that microalgae consume $NO_3$ and $NO_2$ at slower rates compared to $NH_4^+$, as ammonia in contrast to nitrates and nitrites can be included directly into the composition of amino acids, which is necessary for microalgae metabolic functioning and growth, whereas specific enzymes nitrate reductase and nitrite reductase should firstly reduce $NO_3$ and $NO_2$ to ammonium in order for them to later be utilized by microalgae. Moreover, being an energy-dependent process, $NO_3$ transportation to the cell membrane requires direct consumption of ATP [43].

Phosphorus removal from municipal wastewater is possible through chemical precipitation at a pH higher than 8.6 or through microalgae assimilation of phosphorus from wastewater. Due to the fact that pH increased up to 9.4 in the main experiment, both possibilities are relevant in the conducted experiment. Other studies achieved a 55% phosphorus removal rate from agroindustrial wastewater using *C. vulgaris* and *Scenedesmus dimorphus* [35]. In the study conducted by Choi [44], 90.84% of phosphorus was removed from municipal wastewater in the microalgae membrane bioreactor. Removal rates obtained in the current study are average; however, phosphorus removal rates highly depend on the initial nutrient concentration, light, pH, and microalgal species or strain.

As can be seen from the data, the concentration of Na, K, and Zn increased by the end of the wastewater treatment process. A steep increase in Na concentration from 3.24 to 26.20 is explained by the addition of NaOH at the beginning of the main experiment in order to adjust the pH value to the microalgae growth optimum. The K and Zn elevation

might be a result of water evaporation, as microalgae did not absorb K and Zn after the total volume reduction elevation took place. Since during the experiment aeration was used, it could cause mechanical disturbance and breakup of the particles, leading to the release of metal ions [45]. Additionally, some bacteria are able to accumulate metals while consuming organic matter [46]. During wastewater treatment, some of these bacteria die, which also causes the release of ions. The same trend of the increasing concentration of Zn, Cu, Ni, and As during the microalgae wastewater treatment process was reported by Krustok et al. [47]. Microalgae are known for their abilities of metal removal through various mechanisms, such as physical adsorption, complexation, precipitation, chelation, and ion exchange with the help of metallothioneins, crystallization on the cell surface, and chemisorption [45,48,49]. Additionally, removal of metals strictly depends on biotic factors such as microalgal species, tolerance capacity, biomass concentration, and bacterial abundance, as well as abiotic factors such as pH, temperature, metal ionic strength, salinity, hardness, effect of combined metals and metal speciation, and initial concentration [45]. In the study of Mubashar et al. [50], the removal of Cr, Cd, Cu, and Pb from textile wastewater by *C. vulgaris* was studied. The concentration of all metals was above permissible limits of 1 mg/L Cr, 1 mg/L Cu, 0.5 mg/L Pb, and 0.1 mg/L Cd, so 5%, 10%, and 20% dilutions were made to improve removal of metals. The final removal efficiency for all metals was between 25 and 93% in all dilutions. In the present study, no dilutions were made to the initial wastewater, which can provide a great benefit to potential industrial implementation.

In the present study, Ca, besides having the highest initial concentration in the wastewater, had also the second biggest removal rate of 82.7%. This value is in agreement with a previous study conducted by Wang et al. [51], where 80.2% of Ca was removed by *Chlorella* sp. from reverse osmosis concentrate during 16 days of batch cultivation. It was reported earlier that a pH increase may promote the uptake of heavy metals from the water by microalgae, as with pH elevation, the surface of the microalgae cell becomes negatively charged [52,53]. However, Wang et al. [51] concluded that chemical precipitation has the most significant impact on Ca and Mg removal rates during the cultivation of microalgae, which occurs due to pH increase as a result of biomass growth. Thus, an increased pH is likely to facilitate metal removal in both ways: because of microalgal sorption and chemical precipitation. Moreover, the presence of certain bacteria can enhance metal removal, as it was studied by Mubashar et al. [50] that the addition of *Enterobacter* sp. MN17 to textile wastewater during microalgae cultivation with *C. vulgaris* showed better removal rates of Pb, Cu, Cd, and Cr by decreasing wastewater toxicity and intensifying microalgal growth. A potential explanation for the higher removal efficiency of some metals than others may be that microalgae better utilize these metals to maintain their functions and growth [54].

In previous studies of wastewater treatment with microalgae, COD, BOD, and TOC removal rates varied significantly, depending on the initial quality and type of wastewater, treatment duration, and microalgae species used. Hongyang et al. [8] reported a 77.8% removal of COD after cultivation of *Chlorella pyrenoidosa* in soybean processing wastewater. Travieso et al. [9] reported an 88% removal rate of COD after 190 h of treatment in piggery wastewater using *Chlorella* spp.; however, initial wastewater COD was significantly lower compared to the current study and was composed of 250 mg/L. Usha et al. [10] reported an 82% and 75% removal rate of BOD and COD, respectively, and a 75% reduction in TOC in pulp and paper mill effluent after 28 days of cultivation, with a mixed microalgae culture of two *Scenedesmus* species. One more study obtained 85% and 89% TOC removal in two open photobioreactors treating domestic wastewater with mixed microalgal–bacterial consortium [11].

## 5. Conclusions

The obtained results of using mixotrophic species of *C. vulgaris* showed that this alga is particularly effective in reducing COD, BOD, TOC, nitrogen, and certain metal ions from sugar beet wastewater. COD and BOD removal efficiencies were 93.7% and 98.1%, respectively, and TOC content decreased by 95.7%. Nitrites and nitrates decreased by

96%, while the biggest decrease in metal ions was achieved for Ca and Mn (82.7% and 97.6%, respectively). These findings suggest that one of the main goals for obtaining purified water was achieved. In addition, the advantage of incorporating microalgae such as *C. vulgaris* into wastewater treatment is the generation of $O_2$ through photosynthesis, which was recorded in this research when the level of oxygen increased from 0.007 mg $O_2$/L (0.8%—$O_2$) to 1.68 mg $O_2$/L (19.1%—$O_2$). It can be of particular significance in the case of wastewater discharge into surface freshwaters in terms of improving the ecological status of these recipient aquatic ecosystems. On the other hand, these results of increasing $O_2$ concentration by more than 20% indicate the possibility of significant savings in energy demand of wastewater treatment by using the tested *C. vulgaris*. As the results showed, there were no comparable removal rates between control experiments. Aerobic control showed a significant decrease in COD, BOD, and nitrates, while anaerobic treatment showed a noticeable decline in orthophosphates and suspended solids. These results suggest that bacteria naturally present in wastewater contribute to nutrient removal. The results of this study are promising for the potential environmentally friendly application of *C. vulgaris* in the development of an integrated biorefinery in sugar beet processing plants for improved and cost-effective wastewater treatment. It could also be considered particularly important for a multifaceted approach to managing the environmental sustainability of wastewater bioremediation.

**Author Contributions:** N.K.: investigation, formal analysis, writing—original draft preparation. N.M. conceptualization, methodology. P.D.: formal analysis, investigation. D.B.: formal analysis, investigation. M.B.-T.: methodology, supervision, funding acquisition. J.S.: resources, methodology, supervision. V.P.: formal analysis, investigation. Z.Š.: conceptualization, supervision, resources. A.M. (Anamarija Mandić): supervision, funding acquisition. M.P.: conceptualization, supervision. A.M. (Aleksandra Mišan): conceptualization, methodology, supervision. All authors have read and agreed to the published version of the manuscript.

**Funding:** This research was supported by the Science Fund of the Republic of Serbia, PROMIS call, Project No. 6066881, WasteWaterForce, and by the Ministry of Education, Science and Technological Development of the Republic of Serbia (Contract No. 451-03-68/2022-14/200222 and Grant No. 451-03-68/2022-14/200125).

**Institutional Review Board Statement:** Not applicable.

**Informed Consent Statement:** Not applicable.

**Data Availability Statement:** Not applicable.

**Acknowledgments:** The authors would like to thank the sugar factory in Crvenka and, in particular, Goiko Kljajic, Quality Assurance Manager A.D., for their assistance in the implementation of the project.

**Conflicts of Interest:** The authors declare no conflict of interest.

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
