# Peer review of "Sugar Beet Processing Wastewater Treatment by Microalgae through Biosorption"

_water, doi:10.3390/w14060860_

Round 1
Reviewer 1 Report
This paper attempted to analyze the biosorption method of wastewater treatment using microalgae. The source of wastewater sample was chosen as sugar beet processing plant in this study. The authors conducted systematic experiment for 6 weeks to understand the variation in several water quality parameters in both aerobic and anaerobic condition. However, the paper is poorly written with several unstructured sentences including grammatical errors. For example, section 2.5, line 218-227, line 237-240, etc).
The subsection in this paper is completely mixed without justification. For example, the paper is describing and comparing the preliminary results with the results from the main experiments without presenting the main experiment’s data (line 226, 247, etc).
There are so many short paragraphs being written in this paper (for example, section 2.4). There are many abbreviations using first time without explanation (for example TKN).
The discussion part in this paper is completely missing. Overall, it is missing the rigor of a scientific paper.
Author Response
Dear Reviewer,
Thank you for your comments and suggestions. Please, find attached answers to your questions (in red) and the corrected version of the manuscript.
Best regards,
Nadiia Khakimova

Reviewer 2 Report
- The draft does not state whether the wastewater used was sterilized before the experiment. It can be seen from Figure 2 that bacteria may exist in the reactor. The presence of bacteria can have a significant impact on nutrient removal. Therefore, if bacteria are present, both the conclusions and descriptions of this draft should be altered. It is recommended to supplement experimental data on bacterial community structure.
- Due to the presence of light (natural or artificial) outside the reactor, the microalgae adhered to the walls of the reactor. Therefore, it is inappropriate to be called ‘biofilm-based microalgal cultivation’. Adsorbed microalgae are difficult to harvest and obstruct light exposure.
- Mixotrophic culture was used in this work. What light source was used? What is the light intensity on the outer surface of the reactor?
Author Response

(The authors gave the same response as above.)

Reviewer 3 Report
Manuscript Number: water-1580343
Sugar beet processing wastewater treatment by microalgae through biosorption.
GENERAL COMMENTS
The present work investigates the elimination of contaminants in sugar beet processing wastewater through a treatment with microalgae. I found the work quite interesting, besides having a comfortable reading. I have highly valued the fact of carrying out control tests, something that is not very common when microalgae are used in wastewater treatment.
I think the work is interesting and it should be accepted in the present form.
I only have some recommendation for the authors:
(221) (243) Table 1,2 is repeated, and successive.
(350) (380) Correct the subscript
Finally, I have a question for the authors:
During the control experiments, were no measurements of the progress of the pollutants made?
Author Response

(The authors gave the same response as above.)

Reviewer 4 Report
Dear authors
The manuscript „Sugar beet processing wastewater treatment by microalgae through biosorption“ by Khakimova et al., deals with an interesting topic such as the effect of wastewater treatment on the potential and prospects of biosorption of organic and inorganic compounds from sugar plant wastewater by microalgae.
However, as it stands, the manuscript has some points that should be revised. In particular, the Results and Discussion sections are long and should be rewritten in a clearer and more focused manner so as to highlight the strengths and main findings of the work. The topic and research content of this article have some value, but there are still some problems.

Author Response

(The authors gave the same response as above.)

Round 2
Reviewer 1 Report
The revised version of the paper is in good share and it may be ready for publication.
Author Response
Dear Reviewer,
Thank you for your feedback.
Kind regards,
Nadiia Khakimova
Reviewer 2 Report
OK
Author Response

(The authors gave the same response as above.)

Reviewer 4 Report
Dear authors
I appreciate the changes which were done in this manuscript. But there are some parts which should be improved.
Line 18 - 26 °C is not correct it should be so: 26°C without the gap, check the whole document
Formula for calculation of efficiency RE (%) = 100 – (xf × 1000/ xi)
is not correct. According to this formula, the results in the tables do not fit, it would have to be according to the formula below, which is commonly used to calculation of efficiency.
RE(%)=[(influent -effluent))/influent] x100
Line 203 – what does mean (2)? It is the number of reference? In citation number 2 is not mentioned this calculation
Line 233 – mistake (Table 1Table 1)
Line 261 – mistake (Table 2Table 2).
Line 306 – is missing the number of figure (Table 2, Figure )
Line 297– mistake ... (Figure 4, bError! Reference source not found.).
Line 334 - The results are presented on Figure.. is missing number
Conclusion: I do not think that this part is corrected. It should be improved. Conclusion section is not the place for details about your methodology.
Line 475-481 This paragraph is almost copy from the abstract and in conclusion it is unnecessary.
You should give a summary of main finding and their importance and how your research contributes new understanding of this problematics.
Line 491-494 - remove suggestion or speculations which are not solved in the study and also the future plans.
Reference: should contain names of every authors not only initials of their names (see reference 4, 8 and 12). Reference 4 should be written so:
Vaccari, G.; Tamburini, E.; Sgualdino, G.; Urbaniec, K.; Klemeš, J. Overview of the environmental problems in beet sugar processing: possible solutions .....
Best regards
Author Response

(The authors gave the same response as above.)
